# Development of a Method for the In Vivo Generation of Allogeneic Hearts in Chimeric Mouse Embryos

**DOI:** 10.3390/ijms24021163

**Published:** 2023-01-06

**Authors:** Konstantina-Maria Founta, Magdalini-Ioanna Tourkodimitri, Zoi Kanaki, Sylvia Bisti, Costis Papanayotou

**Affiliations:** 1Biomedical Research Foundation Academy of Athens, 115 27 Athina, Greece; 2Department of Medicine, University of Patras, 265 04 Patras, Greece

**Keywords:** CRISPR-Cas9, induced blastocyst complementation, mouse chimera, in vivo heart generation, allogeneic heart, pluripotent stem cells

## Abstract

Worldwide, there is a great gap between the demand and supply of organs for transplantations. Organs generated from the patients’ cells would not only solve the problem of transplant availability but also overcome the complication of incompatibility and tissue rejection by the host immune system. One of the most promising methods tested for the production of organs in vivo is blastocyst complementation (BC). Regrettably, BC is not suitable for the creation of hearts. We have developed a novel method, induced blastocyst complementation (iBC), to surpass this shortcoming. By applying iBC, we generated chimeric mouse embryos, made up of “host” and “donor” cells. We used a specific cardiac enhancer to drive the expression of the diphtheria toxin gene (*dtA*) in the “host” cells, so that these cells are depleted from the developing hearts, which now consist of “donor” cells. This is a proof-of-concept study, showing that it is possible to produce allogeneic and ultimately, xenogeneic hearts in chimeric organisms. The ultimate goal is to generate, in the future, human hearts in big animals such as pigs, from the patients’ cells, for transplantations. Such a system would generate transplants in a relatively short amount of time, improving the quality of life for countless patients around the world.

## 1. Introduction

Thousands of people worldwide suffering from end-stage heart failure need a heart transplant. Due to limited availability of donated organs, only a small percentage of these patients is offered the life-saving treatment every year. Even when they receive a transplant, patients risk severe complications; as the grafted heart originates from another organism, the recipient’s immune system attempts to reject it. Patients are typically put, for the rest of their lives, on immunosuppressive drugs, which cause unwanted side effects, such as an increased likelihood of infections or the development of certain cancers [1].

Stem cell-related technologies promise to generate organs from patients’ cells. Adult cells can be reprogrammed into induced pluripotent stem cells (iPSC) [2,3]. These constitute an extensive source of a starting material which is able to differentiate into any tissue. Moreover, being autologous, they bypass the problem of incompatibility and rejection of the graft by the host immune system. To this end, iPSCs have already been used successfully in animal models of diabetes, liver injury, myocardial infarction and Parkinson’s disease [4,5,6,7]. 

However, generating whole organs for transplantations has proven a more challenging task. Organs are complex structures. Their development in the embryo relies on a series of inductive interactions between different tissues which cannot be easily recapitulated in vitro. Furthermore, organs usually consist of multiple cell types. Therefore, it is not surprising that efforts to populate synthetic or decellularized organ scaffolds with differentiated stem cells, in order to generate functional organs, have not yet met with great success [8,9]. 

The fact that the living embryo is the only system where functional organs can develop successfully led to the idea that such organs could be produced in the context of embryonic development by means of blastocyst complementation (BC). This principle was first demonstrated when wild type mouse embryonic stem cells (mESC) were injected into mutant blastocysts that would develop into mice unable to produce T and B lymphocytes. The mutant host blastocysts provided a “developmental niche” that was occupied by the wild type ESCs, which developed successfully into lymphocytes. All B and T lymphocytes were exclusively derived from “donor” cells [10]. It was subsequently demonstrated that this method could be applied in the case of complex three-dimensional organs. Wild type mouse ESCs or iPSCs were injected into *Pdx1* mutant blastocysts which developed into pancreas-deficient mice. Both cell types were able to colonize the developmental niche and produce functional pancreata, rescuing the lethal phenotype of the host mutants. The developed pancreata were derived almost exclusively from the wild type “donor” cells. It was also shown that mouse *Pdx1* mutant blastocysts could be complemented by rat ESCs or iPSCs. The resulting mice had a functional pancreas consisting almost exclusively of rat cells confirming that iPSC-derived organs can be generated in an xenogeneic environment and paving the way for the production of organs by patient-derived iPSCs in large host species such as swine [11]. The same group then succeeded in using BC to generate allogeneic pancreata in apancreatic pigs, demonstrating that this principle can also be applied to large animals [12]. Ensuing studies resulted in the production of various organs using BC in rodent and non-rodent species alike: thymus, lungs, kidneys, liver, eyes, inner ear and gametes [13,14,15,16,17,18,19,20,21,22,23,24,25].

Promising as this method appears, blastocyst complementation is not suitable for the production of hearts. Attempts have been made to generate rat hearts in mouse embryos, using mouse blastocysts, mutant for the early cardiac marker *Nkx2.5* [26]. However, *Nkx2.5* mutants form a heart, albeit malformed and dysfunctional [27,28]. Accordingly, the developmental niche created in this way contained mouse “host” cells, and the hearts of the resulting chimeras were not exclusively formed by but rather enriched in rat cells. Moreover, the chimeric embryos did not survive to term [26]. As there is no mutation with a heart agenesis phenotype, hearts cannot be generated by means of BC. 

To address this issue, we developed an induced blastocyst complementation approach (iBC). In this, a developmental niche is created by expressing the diphtheria toxin (dtA) in the morphogenetic field of a particular organ, thus killing all “host” cells that give rise to that organ and allowing its formation exclusively by “donor” cells. dtA expression is regulated by a gene enhancer, specifically active in that field. In this work, two mES cell lines were generated. “Donor” cells are tagged with a red fluorescent protein and colonize the entire chimeric body, while “host” cells, tagged with a green fluorescent protein, are excluded from the heart. To achieve this, “host” cells express dtA under the control of the AR1 cardiac enhancer of the *Nkx2.5* marker. *Nkx2.5* is expressed in the first and second heart fields as of day 7.5 of embryonic development (E7.5) [29,30]. It is regulated by different elements, of which the AR1 enhancer is responsible for its early expression in both heart fields [31]. As a result, starting from E7.5, “host” cells in the developing heart will produce dtA and die, creating a heart niche and allowing “donor” cells to occupy it entirely and form the organ by themselves.

Genetic modifications were introduced using CRISPR/Cas9 mediated genome editing in the Hipp11 genomic locus. This site, located between genes *Drg1* and *Eif4enif1* on chromosome 11, is considered a “safe harbor”, allowing the predicable integration of transgenes into the genome without perturbing endogenous gene activity and interfering with the viability and fertility of the organism. Moreover, the expression of transgenes inserted into *Hipp11* is robust and ubiquitous and is not affected by neighboring regulatory elements [32,33,34].

Chimeric embryos were generated using the aggregation of “host” and “donor” mES cells with tetraploid embryos [35]. Tetraploid cells do not contribute to the embryo proper but produce only the extraembryonic tissues necessary for embryo survival. As a result, embryos generated by iBC consist of “host” and “donor” cells and, therefore, show both green and red fluorescence, whereas their hearts are composed of “donor” cells and red fluoresce. 

This is a proof-of-concept study, showing that a heart niche can be formed in the developing embryo and paves the way for a generation of allogeneic and xenogeneic hearts in chimeric organisms. Moreover, by selecting suitable enhancers, this method could be used for the in vivo formation of other organs as well. We believe that iBC will turn out to be instrumental in the efforts made by numerous labs worldwide to produce organs suitable for transplantation.

## 2. Results

### 2.1. Generation of “Donor” Cells

In order to integrate a tdTomato-expressing cassette into the mouse genome, first the online tool E-CRISPR (http://www.e-crisp.org/E-CRISP/, accessed on 10 November 2016) was used to identify Cas9 targets in the Hipp11 locus. The sequence 5′-GATGTGAACAAAGCACCCTA-TGG-3′ was chosen as it produces only one hit with a high predicted efficiency (Xu score: 0.59) and is located in the middle of the Hipp11 sequence. Subsequently, a plasmid expressing both Cas9 and the guide RNA (gRNA) was constructed. A “donor” plasmid was also constructed containing the gene of the tdTomato under the control of the strong, constitutive promoter CAG and the neomycin resistance cassette. These two elements were flanked by left and right homology arms (LHA and RHA, both ~900 bp), converging in the point of the Cas9-induced double strand break. Two versions of this “donor” plasmid were made, one of which also contained Cas9 target sequences, permitting the excision of the introduced elements from the plasmid backbone (Figure 1A). Each “donor” plasmid was transfected into mESC together with the Cas9-gRNA plasmid. Transfection was followed by G418 selection. The incorporation of the Cas9 targets into the “donor” plasmid greatly enhanced the efficiency of the procedure as it resulted in a mean of 12 fluorescent colonies per transfection as opposed to 9 colonies, of which only 7 fluoresced when the Cas9 target was missing (n = 3, Figure 1B). Accordingly, all plasmids constructed afterwards included Cas9 target sequences. Stably transfected cells were expanded and showed a high fluorescence signal, undiminished over multiple cell passages (Figure 1C). The “Donor” cells were subsequently subcloned. In total, 16 clones were expanded and analyzed. They all fluoresced red. The PCR analysis on genomic DNA, using an external and internal primer, showed that all 16 clones incorporated the expression cassettes in the correct location (Figure 1D). A second pair of primers revealed that integration had taken place in one of the two alleles in all examined clones (Figure 1E) and all clones were, therefore, heterozygous.

To assess possible off-target events of the CRISPR-mediated *tdTomato* integration, the online tool CRISPOR was used (http://crispor.tefor.net, accessed on 12 September 2021). In total, 154 such events were predicted. The three most likely to occur (with a cutting frequency determination of > 0.5) were evaluated. In addition to these, the 13th most likely off-target event overall (with CFD = 0.27), but first inside an exon (of the gene *Ncapd3* in chromosome 9), was also assessed. In total, ~500 bp fragments of genomic DNA from transfected cells containing the putative off-targets were PCR amplified and sequenced. No such genome editing events were identified (Appendix A). 

In short, a method was developed to introduce CRISPR/Cas9 into the Hipp11 “safe harbor” locus of mESC, a fluorescent protein expression cassette, and generate the “donor” cell line. Knock-in was fast, efficient and accurate and produced no off-target genome editing events. 

### 2.2. Study of the Activity of Nkx2.5 AR1 Cardiac Enhancer in Hipp11

#### 2.2.1. AR1 Upregulates EGFP Expression in Differentiating Cardiac Cells 

For iBC to succeed, the enhancers of the genes involved in the early steps of organogenesis must be used. To create a heart niche, the selected enhancer must be inactive in undifferentiated cells and become active upon their differentiation into cardiac progenitor cells. The AR1 enhancer of the *Nkx2.5* cardiac marker is reportedly activated in the first and second heart fields between 7.5 and 8.0 days of embryonic development (E7.5–8.0) [31].

In order to evaluate its activity in the Hipp11 (where it will regulate the expression of dtA in “host” cells), a cassette expressing EGFP under the control of AR1 was introduced into the genomic locus using CRISPR/Cas9. To this end, a plasmid was constructed containing a neomycin resistance cassette and EGFP under the control of the minimal promoter E1b [36] and enhancer AR1, flanked by LHA, RHA and Cas9 target sequences (Figure 2A). A second control plasmid, lacking the enhancer, was also made. The two plasmids were co-transfected into the mESC with the Cas9-gRNA plasmid. Stable transfectants formed embryoid bodies (EB) that were either cultured for 12 days or plated on day 4 and allowed to differentiate in vitro for 8 more days in order to give rise to beating cardiomyocytes. Day 12 (d12) AR1-EGFP cells show a marked increase in EGFP expression compared to control cells (Figure 2B). Individual d12 green fluorescent cells also express cardiac troponin T (cTnT) (Figure 2C). The time-course fluorescence study of developing EB indicates that EGPF starts being expressed on day 4. The expression is strong and widespread already by day 6. By day 8, fluorescent cells start migrating, and on day 12, they occupy the center of the EB, as do all the cells of mesodermal origin (Figure 2D–I). These results suggest that the AR1 cardiac enhancer, inserted in Hipp11, is inactive in undifferentiated mESC but is activated during their differentiation into cardiomyocytes.

#### 2.2.2. AR1 Upregulates Flp Expression in Differentiating Cardiac Cells

EGFP is not an overly sensitive read-out of AR1 activity. It is conceivable that AR1 in mESC may not be active enough to provide a detectable fluorescent signal but can produce amounts of dtA capable of killing the cells. Enzyme function is much more likely to be affected by enhancer activation, as a few molecules are typically sufficient to catalyze a reaction. For this reason, and in order to confirm the suitability of AR1 for our purposes, an additional assay was developed, using flippase (Flp) instead of EGFP. To that end, another plasmid expressing EGFP under the control of CAG was constructed. The EGFP gene was flanked by two FRT elements and followed by the tdTomato gene, which is not expressed, as it is separated from the EGFP by nine stop codons (all three of them in all three open reading frames) and two polyadenylation signals. The plasmid also contains the neomycin resistance cassette and an attP element. All these sequences are flanked by the left and right HA and two copies of the Cas9 target (Figure 3A). mESC were co-transfected with the EGFP-tdTom and Cas9-gRNA plasmids. Stable transfectants, selected using G418, contained the EGFP-tdTom cassette in the Hipp11 locus. The EGFP-tdTom cells were subjected to a second round of transfection using a plasmid expressing flippase under the control of the minimal promoter E1b and AR1 enhancer and containing a puromycin resistance cassette and attB element (Figure 3A). This plasmid was co-transfected with a second one, expressing the φC31 integrase under the control of the strong, constitutive promoter CAG. φC31 catalyzes the irreversible recombination between the attP and attB elements, resulting in the incorporation of E1b-Flp-AR1 into Hipp11. The stable EGFP-tdTom/Flp-AR1 transfectants, selected using puromycin, emitted green but not red fluorescent signals (Figure 3B). When these cells formed EB, they started showing red fluorescence after 4 days. Evidently, at this point, AR1 started being activated in differentiating cardiac cells and induced the expression of Flp. The enzyme catalyzed the recombination between the two FRT elements, removing the EGFP and bringing the CAG promoter directly upstream of the tdTomato gene, which was consequently expressed at high levels. By day 6 of EB formation, the red fluorescent signal was extensive and strong (Figure 3C), while on day 12, red fluorescent cardiomyocytes were localized in the center of the EB, and green fluorescent cells occupied their surface (Figure 3D). These results show unequivocally that the AR1 cardiac enhancer is not active in mESC and that it is activated in differentiating cardiac cells. 

### 2.3. Generation of “Host” Cells

In order to create a heart niche in the developing embryo, green fluorescent “host” cells need to die in the first and second heart fields at the time of their specification. Their place will be occupied by “donor” cells, giving rise to an exclusively red fluorescent heart. The death of the “host” cells in the developing heart will be mediated by dtA, whose expression will be regulated by the AR1 cardiac enhancer. To generate “host” cells, another plasmid was made. It contains an EGFP cassette, expressing the fluorescent protein under the control of the CAG promoter, a neomycin selection cassette and a third cassette, expressing dtA under the control of the E1b minimal promoter and AR1 enhancer (Figure 4A). Once again, these elements are flanked by LHA, RHA and two Cas9 target sequences to allow CRISPR/Cas9-mediated integration into the Hipp11 genomic locus. mESC were transfected with this plasmid, and stable transfectants were selected using G418. “Host” cells fluoresce green and can be typically expanded (Figure 4B). This suggests that, in undifferentiated cells, AR1 in the Hipp11 locus is not active, and dtA is not produced. “Host” cells were subsequently differentiated into cardiomyocytes, and the expression of the *Nkx2.5* cardiac marker in these cells was assessed using real time RT-PCR. “Host” cells showed significantly lower levels of *Nkx2.5* expression compared to differentiated control cells, suggesting that upon their differentiation into cardiac cells, they upregulate dtA and die. In conclusion, we succeeded in generating green fluorescent “host” ES cells that, upon differentiation along the cardiac cell line, express dtA and die.

### 2.4. Generation of Chimeric Embryos

Having verified the suitability of the “host” cell line, we proceeded with the generation of chimeric embryos, consisting of both “host” and “donor” cells. To do so, tetraploid aggregation chimera formation was performed. Embryos at the two-cell stage were electrofused to give single tetraploid cells. These were allowed to grow into four-cell stage tetraploid embryos. Two such embryos were aggregated with a mixture of approximately 20 “host” and “donor” cells at a 1:1 ratio. The aggregates were left to develop into blastocysts, which were subsequently implanted into the uterus of pseudopregnant female mice. The foster mothers were sacrificed when the embryos reached stage E9.5 of embryonic development. The embryos are composed exclusively of “host” and “donor” cells, as tetraploid cells of the aggregated embryos do not contribute to the embryo proper but are restricted to extra-embryonic lineages. Three such embryos were collected; they all showed the same phenotype. The embryonic body consisted of a mixture of green “host” and red “donor” cells, except for the heart tube, which showed a striking absence of green fluorescence. In contrast, the developing heart showed a red fluorescent signal which was significantly more intense than in any other parts of the embryo, suggesting that the “donor” cells had replaced the “host” cells and constituted the sole cell population in that part of the embryo. Interestingly, the same pattern was observed in an area of the branchial arches, known to originate from the second heart field and give rise to facial muscles (Figure 5). This result suggests that it is possible to generate embryos with a developmental heart niche, that can be colonized by donor cells, for an allogeneic organ to be created.

## 3. Discussion

In this study, a novel CRISPR/Cas9 target site in the Hipp11 locus of the mouse genome was identified. Hipp11 is an intergenic region in the mouse chromosome 11, located between the genes *Drg1* and *Eif4enif1.* It is considered a “safe harbor” for the insertion of transgenes. DNA sequences introduced in Hipp11 do not interfere with endogenous gene activity. Moreover, the viability and fertility of the genetically engineered mice is not affected. More importantly, neighboring regulatory elements do not have an impact on the expression of the integrated transgenes [32,33,34]. The new CRISPR/Cas9 target sequence is unique in the mouse genome and predicted to be efficiently cleaved by Cas9. Indeed, we found that CRISPR/Cas9 allows efficient and accurate incorporation of a fluorescent protein expression cassette in the targeted site using homologous recombination. All clones examined contained the *tdTomato* gene, expressed under the control of the strong promoter CAG, in the correct position and in one of the two alleles. Our results prove that this is an adequate system to tag cells, allowing robust and ubiquitous monoallelic expression of a gene marker. It also permits a comparison between different cells lines, as the expression of the marker remains, in theory, stable and equivalent. 

We also inserted in the same position of Hipp11, another gene cassette expressing *EGFP* under the control of the minimal promoter E1b and the cardiac enhancer AR1 of the *Nkx2.5* cardiac marker. The transfected cells expressed the fluorescent marker as was expected. Undifferentiated cells showed no EGFP expression; instead, the fluorescent protein started being synthesized upon the differentiation of the stem cells along the cardiac lineage. This confirms that E1b cannot initiate transcription by itself and can be unhesitatingly used in such types of assays. It also validates our system as a dependable means to study enhancers: as long as there exists a reliable method for differentiating stem cells into specific cells types, the function of the regulatory elements, active during the differentiation process, can be comprehensively investigated. Additionally, real-time RT-PCR provides a means for the quantitative analysis of the activity of such enhancers. Hipp11 has been already used for the study of enhancer activity [33]; however, to our knowledge, this is the first time that CRISPR has been used for that purpose, rendering this type of study faster and easier to perform.

We then developed a system, where flippase, expressed under the control of the AR1 cardiac enhancer, acts as a switch between two different fluorescent proteins, changing the color of the fluorescence emitted by the cells that differentiate along the cardiac lineage. Such a system can be introduced into Hipp11 in two rounds of transfection. In the first round, CRISPR/Cas9 is used to insert the fluorescent marker genes under the control of the strong, constitutive CAG promoter into the genomic locus. In the second round, φC31 integrate incorporates the AR1-*Flp* module into the same area. As Flp is an enzyme, this system is sensitive to even small increases in enhancer activity and, as a result, complements nicely the approach described previously. In theory, the AR1-*Flp* moiety could be replaced by a conditional Cre cassette, changing the switch from a developmental state to time responsive.

The main goal of this study was the development of a method for the generation of allogeneic and xenogeneic hearts in chimeric mice. Over the last decade, many labs have been using blastocyst complementation to create various organs in vivo [37]. This method relies on mutations with specific organ agenesis phenotypes. In such mutant embryos, a developmental niche for the particular organs is created and is occupied by “donor” cells giving rise to the missing organs. Unfortunately, there exists no such mutation for the heart. Mutations in single genes that play seminal roles in heart development always cause the formation of a rudimentary and malfunctioning organ, as exemplified by the *Nkx2.5* mutant mice [26]. As a result, blastocyst complementation experiments give rise to hearts enriched in “donor” cells and never exclusively formed by them. This may be due to the redundancy between the genes involved in cardiogenesis. Indeed, the only known way to completely abolish heart formation in the developing embryo is to silence both *Mesp1* and *Mesp2* [38]. Nevertheless, this double mutation affects not only the heart but all posterior structures and is, therefore, so devastating to the embryo, as to make it impractical for blastocyst complementation assays. 

In order to surpass this limitation, we developed a novel method, which we termed “induced Blastocyst Complementation” (iBC). According to this method, an empty developmental niche is formed by means of the expression of the diphtheria toxin dtA. The gene of the toxin is regulated by an enhancer, which is active in the equivalent morphogenetic field. The developmental niche can be completely occupied by “donor” cells, which form the corresponding organ with no contribution by the “host” cells, as these express dtA and die when they differentiate into the undesired cell types.

To this end, we used the AR1 cardiac enhancer of *Nkx2.5.* The cardiac marker is expressed from day 7.5 of embryonic development (E7.5) in the first and second heart fields [31], which give rise to almost all heart tissues [39,40,41]. Expression of the gene is initiated in the prospective cardiac cells by AR1. Therefore, the cardiac enhancer is activated in almost all the cells that will contribute to the heart and can be used to create a developmental niche, ready to be occupied by “donor” cells. In order to generate the “host” mES cell line that is excluded from the developing heart, a cassette expressing dtA under the control of the E1b promoter and AR1 enhancer was inserted into Hipp11. The fact that these cells can be maintained in culture confirms once again that E1b is, indeed, a minimal promoter, unable to initiate transcription by itself but also suggests that AR1 is not active in undifferentiated cells. These cells, when subjected to a well-established differentiation protocol, fail to upregulate *Nkx2.5*, suggesting that they express dtA and die upon their differentiation along the cardiac lineage. This result indicates that “host” cells behave in vitro as expected and can be used in in vivo experiments. 

Finally, we proceeded with the generation of chimeric embryos, composed of green fluorescent “host” cells prone to die in the heart field and red fluorescent “donor” cells. To do so, the tetraploid aggregation method was selected: “host” and “donor” cells were incorporated into the tetraploid blastocysts. Tetraploid cells are excluded from the embryo proper and contribute only to extraembryonic tissues. As a result, the chimeric embryo consists exclusively of the supplied stem cells [35]. We managed to collect three embryos at stage E9.5. They consist of green fluorescent “host” and red fluorescent “donor” cells. Interestingly, the heart tubes show a marked absence of green fluorescence. In contrast, red fluorescence in the prospective heart is more intense than in other parts of the embryo, suggesting that “donor” cells had replaced the missing “host” cells. Significantly, the same pattern is discernible in parts of the head that are known to originate from the second heart field [42,43]. Evidently, in these cells not only is *Nkx2.5* expressed but also this expression is regulated by the AR1 enhancer. 

To our knowledge, this is the first time in more than a decade of blastocyst complementation experiments that allogeneic hearts are being generated in developing mouse embryos. There is only one such attempt reported in the literature, and this resulted in chimeric hearts in embryos that did not survive to term. The reason for this chimerism is the lack of suitable mutations with a heart agenesis phenotype. To overcome this limitation, we developed a novel method, which we termed induced blastocyst complementation and which promises to create hearts in host embryos by expressing the diphtheria toxin under the control of a specific cardiac enhancer in order to empty the corresponding developmental niche of host cells. 

We are now developing a system to remove unnecessary sequences from the Hipp11, in particular the neomycin selection cassette and AR1-*dtA* cassette, after “host” cell death in the heart fields has taken place between E7.5 and E8.5, leaving only the fluorescent cassettes to discriminate red fluorescent “donor” from green fluorescent “host” cells. Once this system is in place, chimeric embryos will be allowed to develop to term. Adult mice will be sacrificed, and the contribution of the “host” and “donor” cells in different organs will be assessed using fluorescence and real-time RT-PCR. We expect that all organs will have been formed by a mixture of the two cell types, with the exception of the heart which we believe will consist exclusively of “donor” cells. We are also developing a system to exclude “donor” cells from the remaining body to produce a chimeric animal with a “donor”-derived heart in a “host”-derived body. This will ensure that “donor” cells will not colonize tissues such as the brain and germ line, which is an outcome that poses severe bioethical issues. Finally, we plan to apply iBC to the production of xenogeneic hearts in interspecies chimeras.

## 4. Materials and Methods

### 4.1. mESC Culture

CK35 mESC (a kind gift from Dr Francina Langa) were cultured on mitomycin-treated SNL feeders plated on gelatin (Sigma, St. Louis, MO, USA), in Dulbecco’s Modified Eagle Medium (DMEM, Gibco, Waltham, MA, USA) supplemented with 15% heat-inactivated ES-qualified fetal bovine serum (FBS, Biosera, Hoddesdon, UK), 2 mM glutamax (Gibco) and 0.1 mM β-mercaptoethanol (Sigma) at 37 °C and in 5% CO_2_. Cells were passaged every 2 days using 0.05% trypsin (Gibco).

### 4.2. mESC Transfection

mESC were transfected with lipofectamin 3000 (Invitrogen, Waltham, MA, USA). Briefly, 200,000 cells were plated in a 24-well-plate well. Next, 1 μg of total DNA (0.5 μg of each of two plasmids) was combined with 1 μL of 3000 reagent in 25 μL DMEM, combined with a 2 μL of lipofectamin in 25 μL DMEM and added in the cell culture. After 24 h, selection was carried out using G418 (Applichem—0.2 mg/mL for 7 days) or puromycin (Applichem—1.5 μg/mL for 4 days).

### 4.3. mESC Differentiation

In total, 10,000 mES cells were transferred in a 96-U bottom-well-plate well in differentiation medium (DMEM supplemented with 15% heat-inactivated FBS, 2 mM glutamax (Gibco), 0.1 mM β-mercaptoethanol and 100 U/mL penicillin/streptomycin) and cultured at 37 °C and in 5% CO_2_ for up to 12 days for embryo body formation or for 4 days, followed by plating in a gelatinized 12-well-plate well and culturing for an additional 8 days until beating cardiomyocytes appeared. 

### 4.4. Immunofluorescence

Cells on coverslips were fixed in 4% paraformaldehyde, permeabilized in PBS/0.3% triton blocked with 10% FBS in PBS and incubated with the primary (anti-GFP-FITC goat polyclonal-ABCAM ab6662 and anti-cardiac troponin T [1C11] mouse monoclonal-ABCAM ab8295) and secondary (ALEXA FLUOR 568 goat anti-mouse ABCAM ab175473) antibodies diluted in blocking solution (1:500). Nuclei were marked using DAPI (Sigma), and the coverslips mounted on slides using Eukit (Sigma).

### 4.5. Imaging

mESC colonies, embryoid bodies and embryos were observed using Leica DMIRE2 inverted fluorescent microscope (10× objective, light source—CoolLed PE300-White, Camera—ORCA CT-sCMOS). Cells mounted on coverslips were observed using Leica DMCA2 upright fluorescent microscope (10× objective, light source—CoolLed PE30-Ultra, camera—ORCA 4-sCMOS).

### 4.6. Real-Time RT-PCR

Total RNA was isolated using Nucleospin RNA (Macherey-Nagel) according to the manufacturer’s instructions. cDNA synthesis was carried out using Primescript (Takara) and random hexamers. Real-time PCR was performed in C1000 Touch Thermal Cycler2 (Bio-Rad, Hercules, CA, USA) using CFX96Real Time System. Primers are listed in Appendix A.

### 4.7. Generation of Aggregation Chimeras

All procedures for care and treatment of mice were approved by the Institutional Committee on Ethics of Animal Experiments and the Greek Ministry of Agriculture. B6D2F2/J mouse zygotes were collected in M2 (Sigma) and cultured in amino acid supplemented KSOM (Millipore, Burlington, MA, USA) at 37 °C and in 5% CO2. When they reached 2-cell stage, they were electrofused in freshly made 0.3 M mannitol (SIGMA) and 0.3% BSA (SIGMA) in H_2_O, with two pulses of 30 V, 40 μs long each, using the CF-150/C electroporator (BLS) and GSS-250 electrode. They were subsequently cultured in KSOM + AA until 4-cell stage and shelled in Tyrode’s solution (Sigma). Two such embryos were co-cultured in KSOM + AA with groups of ~20 “host” and “donor” mES cells at a 1:1 ration until they reached the late blastocyst stage. At that point, they were transferred into the uterus (up to 10 blastocysts) of E2.5 pseudopregnant mice, where they developed until stages E9.5 and E10.5. For a detailed protocol, see [35].

## 5. Conclusions

We have developed a novel method, induced blastocyst complementation, to generate in vivo allogeneic and xenogeneic hearts. Conceivably, this technology could be transferred to big livestock animals in order to produce hearts from human iPSCs for transplantation. With minor modifications, this method could also be used for the generation of other autologous organs, benefiting numerous patients worldwide in need of transplants.

## Figures and Tables

**Figure 1 ijms-24-01163-f001:**
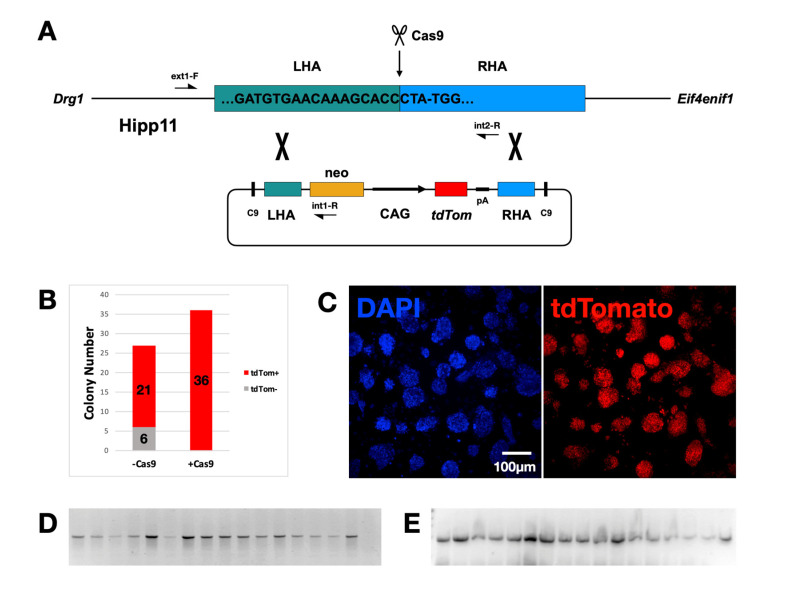
Generation of the “donor” ES cell line. (**A**) Schematic of the homologous recombination between the mouse Hipp11 genomic locus and the “donor” plasmid. Cas9 cleaves the target sequence in Hipp11, at a site 3 nucleotides upstream of the protospacer adjacent motif (PAM) -TGG, separating the left and right homology arms (LHA and RHA). The “donor” plasmid contains the neomycin resistance cassette (neo) and the tdTomato expression cassette (CAG promoter, tdTom gene and polyadenylation signal pA), flanked by the homology arms, which direct the homologous recombination. Two CRISPR/Cas9 target sites (C9), identical to the one in Hipp11, render the recombination event more efficient. (**B**) Bar graph showing the number of stably transfected mouse embryonic stem cell (mESC) colonies with and without fluorescence, in the presence and absence of CRISPR/Cas9 target sites (C9) in the “donor” plasmid (n = 3). (**C**) Stably transfected “donor” colonies showing red fluorescence. (**D**) Gel image showing genomic PCR analysis of 16 “donor” clones, using primers ext1-F/int1-R (indicated in Figure 1A). All clones have incorporated the plasmid in the correct position. Last lane: genomic DNA of cells, transfected without the Cas9/gRNA plasmid (negative control). (**E**) Gel image showing genomic PCR analysis of the 16 “donor” clones in Figure 1D, using primers ext1-F/int2-R (indicated in Figure 1A). All clones have incorporated the plasmid in one of the two alleles. Last lane: genomic DNA of untransfected cells (positive control). Primer sequences are listed in the Appendix A.

**Figure 2 ijms-24-01163-f002:**
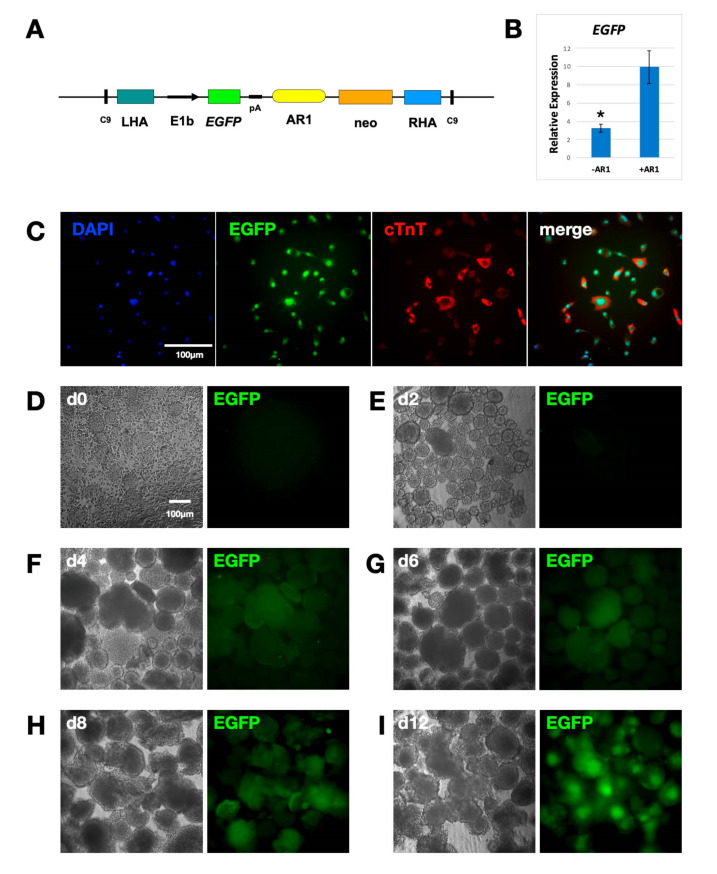
The AR1 cardiac enhancer in Hipp11 activates expression of EGFP in differentiating cells. (**A**) Schematic of the AR1-EGFP plasmid. It contains the neomycin resistance cassette (neo) and the EGFP gene under the control of the minimal promoter E1b and AR1 enhancer, flanked by the LHA, RHA and two CRISPR/Cas9 target sites (C9). (**B**) Real-time RT-PCR analysis showing relative expression of EGFP in the Hipp11 locus of differentiated cells, in the absence and presence of AR1 (n = 3, * *p* < 0.05). Primers are listed in the Appendix A. (**C**) Immunofluorescence of AR1-EGFP mES cells, differentiated into cardiomyocytes. Cells expressing cardiac troponin T (cTnT) also upregulate EGFP. (**D**) Colonies of undifferentiated AR1-EGFP mESC showing no fluorescence. Magnification 10×. (**E**–**I**) AR1-EGFP embryoid bodies (EB), on days 2, 4, 6, 8 and 12, showing upregulation of EGFP, starting from day 4 of EB formation. On day 8, EGFP-expressing cardiac cells start migrating into the centre of the EB. Magnification 10×.

**Figure 3 ijms-24-01163-f003:**
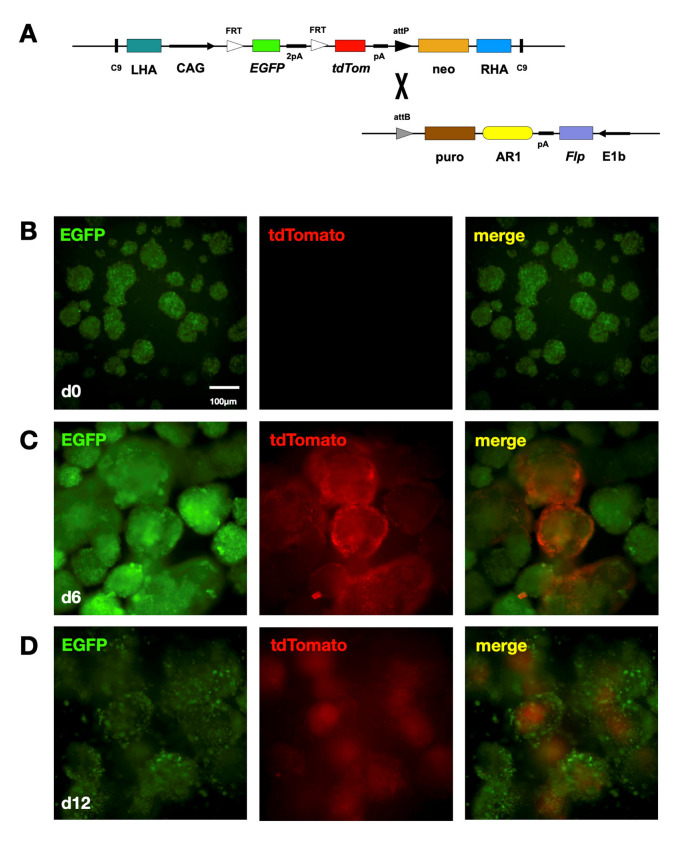
The AR1 cardiac enhancer in Hipp11 activates expression of flippase in differentiating cells. (**A**) Schematic of the EGFP-tdTom and Fpl-AR1 plasmids. The former contains the neomycin resistance cassette (neo) and fluorescent protein-expressing cassette consisting of the EGFP gene and 2 polyadenylation signals (pA) between 2 FRT elements, under the control of the CAG promoter and followed by tdTom. The 2 cassettes, together with an attP element, are flanked by LHA, RHA and two CRISPR/Cas9 target sites (C9). The latter contains an attB element, the puromycin resistance cassette (puro) and flippase (Flp), under the control of the E1b minimal promoter and AR1 enhancer. φC31 integrase catalyzes the irreversible recombination between attP in Hipp11 and attB in the Fpl-AR1 plasmid, resulting in the insertion of the latter into the former. (**B**) Colonies of undifferentiated EGFP-tdTom/Flp-AR1 mESC showing green fluorescence only. Magnification 10×. (**C**,**D**) EGFP-tdTom/Flp-AR1 embryoid bodies (EB), on days 6 and 12, showing tdTomato expression. On day 6, red fluorescence is strong and widespread. On day 12, tdTomato-expressing cardiac cells occupy the centre of the EB, while EGFP-expressing non-cardiac cells are seen on the periphery. Magnification 10×.

**Figure 4 ijms-24-01163-f004:**
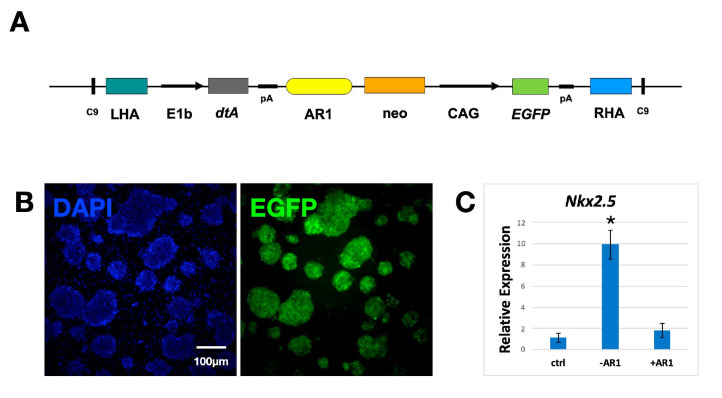
Generation of the “host” ES cell line. (**A**) Schematic of the “host” plasmid. It contains the diphtheria toxin gene *dtA,* under the control of the minimal promoter E1b and AR1 enhancer, the neomycin resistance cassette (neo) and EGFP gene under the control of the CAG promoter. The 3 cassettes are flanked by LHA, RHA and two CRISPR/Cas9 target sites (C9). (**B**) Stably transfected “host” colonies showing green fluorescence. (**C**) Real-time RT-PCR analysis showing relative expression of *Nkx2.5* in differentiated “host” cells, in the absence and presence of AR1 (n = 3, * *p* < 0.05). In the presence of the enhancer, cardiac cells upregulate *dtA* and die, resulting in a marked drop in *Nkx2.5* expression compared to control cells (without AR1). Ctrls are untransfected, undifferentiated mESC. Primers are listed in the Appendix A.

**Figure 5 ijms-24-01163-f005:**
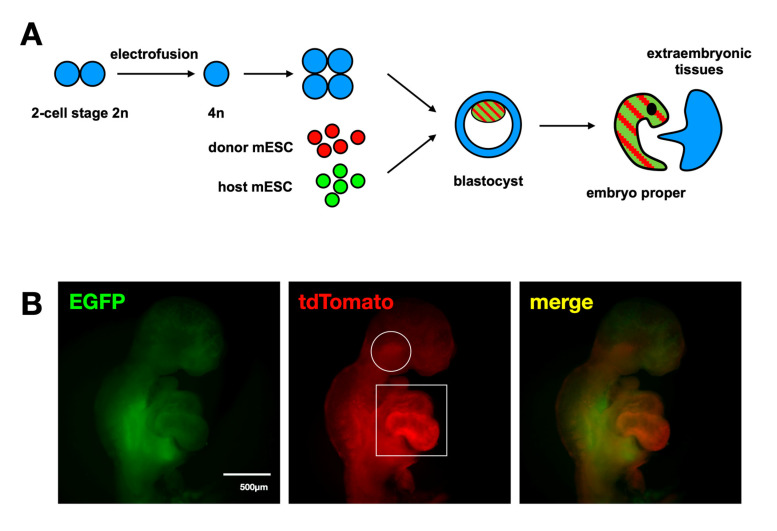
Generation of “host”–“donor” chimeric embryos. (**A**) Schematic describing the generation of chimeric embryos by aggregation of mESC with tetraploid embryos. (**B**) Chimeric embryo showing absence of green fluorescence and increased red fluorescence in the heart tube and pharyngeal arches. Both areas of the embryo originate from the heart fields, which express *Nkx2.5*.

## Data Availability

Not applicable.

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
