# Peer review of "Development of a Method for the In Vivo Generation of Allogeneic Hearts in Chimeric Mouse Embryos"

_ijms, 2023, doi:10.3390/ijms24021163_

Round 1
Reviewer 1 Report
Please detail the experiments in the abstract.
Some minor fixes:
43 However, generating whole organs for transplantations, has proven a more challenging task. (delete comma)
298 Three such embryos were collected. ,tThey all showed the same phenotype.
replace point by comma
469 For a detailed protocol see: Manipulating the Mouse Embryo – Chapter 12. (Authors??)
Author Response
We thank reviewer #1 for embracing our work. We have included the suggested corrections in the revised manuscript.
Reviewer 2 Report
The Authors test whether heart organ production is possible from donor cells exclusively in a chimeric embryo. The idea will be of high value if applicable as organs for transplantation will be possible to be produced in animals like pigs. However, there is a number of experiments that is required to support the conclusion.
1. The authors should show high magnification of the heart tissue and with amplification using antibodies against GFP, tomato and Cell type markers (CM markers like Mef2C, TroponinT etc) to show that cells are of donor origin. As is it is not obvious that the hearts are composed of only red donor cells.
2. The authors only evaluate at E9.5 but should use older ages to confirm the lack of cardiomyopathy as they themselves notice is a challenge in previous settings.
3. In the discussion, the authors suggest to: “Once this system is in place, chimeric embryos will be allowed to develop to term. Adult mice will be sacrificed and the contribution of “host” and “donor” cells in different organs will be assessed by fluorescence and Real-Time RT-PCR.” To this reviewer it seems as if the study is rather preliminary and should await these experiments to confirm that the methods is working. Until then the method is not really of any value.
Author Response
With all due respect, we disagree with reviewer #2. We believe that our work is of considerable value. In the decade that blastocyst complementation experiments have been carried out, there have been many reports on generation of various organ, most notably the pancreas. However, in the case of the heart, there has been only one published work that essentially describes the inability to generate this organ by means of blastocyst complementation, because of the lack of a heart agenesis mutation. That result was considered significant enough to be published in Cell (Wu et al., 2017). For the first time, and after five years from that report, we propose a different system designed to overcome this limitation of the original method. We believe that our work is both innovative and promising (the results on the embryos corroborating this idea) and that it deserves to be published in IJMS, so that other people in the field learn about it and start using and improving it. We would like to stress that nobody ever mentioned cardiomyopathy anywhere. Wu and al. report that chimeras do not survive to term but do not give any reason why. We agree that higher magnification of the embryonic hearts would be a desirable addition to our results. We did take sections of our embryos, however, neither the EGFP, nor the tdTomato signal survived dehydration of the tissue and there is simply no time to repeat the experiment within the time frame given by the journal for the revisions. Finally, while we agree that live chimeras would be the ultimate confirmation of the validity of our method, from our standpoint such results would be the object of an altogether different publication.
Reviewer 3 Report
The author performs chimeric mice experiment to provide a creative design of complement blastocyst experiment. Using an inducible system using an enhancer to selective induced cell death in the heart development phase, so the donor cell can take place to results in a donor cell heart in a host body.
Throughout examination of the manuscript, I would like to offer the following feedback.
1. The author lacks the proper assay of random integrations and supplied incomplete data for off-target editing.
2. The author lacks a critical control mouse (host cells without AR1 + donor cells) and sufficient quantification of donor/host ratio of cell contributions in existing chimeric mice.
3. No surviving post birth organism with red hearts are reported, which to untrained eyes might be undesired outcomes. The 3 incomplete term embryos perhaps are already significant in this field of research. If the author believes their work bring significant progress to the field. I suggest the author highlight the comparison of their work and others in the field. Currently some levels of this type of discussion exists in the middle of the discussion, I suggest the author highlight their impact at the end of the discussion.
Line 20, the author suggest it is possible to produce allo/xeno heart in chimeric organisms. I suggest to scale back the claim to either add “in future” or “in chimeric embryos or organisms”. As the author did not provide data to support chimeric “organisms” in this report.
Line 63 pdx1 mice work lack proper reference
Line 67 the reference is mice/rat xeno work. For Swine xenotransplantation work, perhaps the author could reference https://www.nejm.org/doi/full/10.1056/NEJMoa2201422 or https://pubmed.ncbi.nlm.nih.gov/34331749/
Fig 1a/b study design using Cas9 site in the donor plasmid is a big risk of generation of linear dsDNA, which erase the benefits of using circular plasmids for HDR process. Linear dsDNA greatly increases the chance of random integration and tandem integration. The author lacks the data to exclude random integration or tandem integrations (nanopore sequencing or equivalent). The ex/int PCR can confirm at least one copy went into the Hipp11 locus, however, the gene could have also been inserted into other regions of the genome or tandemly inserted into Hipp11 locus.
Fig1E the gel is cut off the lower part, where some amount of smaller PCR product can be seen, it indicates the unedited allele based on the primer locations. I suggest the author comments on the ratio of homozygous and heterozygous clones. Is it true all clones generated are homozygous?
Figure S1 is a gel picture of 4 off-target sites PCR products, the sanger sequencing results are not attached. I would like to suggest the author to provide the sequencing results and interpretation.
Fig2C has many cells that are GFP + but does not express troponin. Indicating that those cells either failed to differentiate into heart muscle cells or those heart muscle cells does not expression troponin. I recommend the author to add additional controls to interoperate this result. The risk is the lacky expression pattern of AR1 enhancer. Perhaps it does express in heart development but not limit to these cell types.
Fig 4B is not strong evidence to suggest the AR1-dtA plasmid is harmless to ES cells. typically expanded perhaps need to be quantified using MTT assay or equivalent to show the cell proliferation abilities of AR1-dtA cells vs control.
Fig 4C lack the control of undifferentiated ES cells. The level of ES Nkx2.5 is important to guide the interpretation of the cardiac differentiation.
Fig 5B shows the leaky pattern of red cells, it does not limit to heart development as the author have previously planned, it leaky into other parts of the body and especially to the branchial arches.
1. This could due to the previously observed leaky pattern of AR1 enhancer in Fig 2C. This could mean that host green cells die not only at the heart area during heart development.
2. It is not clear if the author has the proper control chimeric mouse [host cells without the AR1 enhancer from Fig4C + red donor cells = control chimeric].
3. The author also lacks the quantification of cell amounts in different parts of the embryos. The comparison of florescent intensity is great for visual effects. Perhaps the author could use FACS/flow to quantify cell numbers based on their sources of origins. (or sequencing based approach)
The ending paragraph consist of many indeed exciting future directions. However, those have no data or evidence to support. I would like to suggest the author to reduce the amounts of future plans. Perhaps only keep relevant descriptions of the impact of the current work, which are supported by data and discovery reported in this current study.
